# Is Pre-Exposure Prophylaxis a Cost-Effective Intervention to Avert Rabies Deaths among School-Aged Children in India? Comment on Royal et al. A Cost-Effectiveness Analysis of Pre-Exposure Prophylaxis to Avert Rabies Deaths in School-Aged Children in India. *Vaccines* 2023, *11*, 88

**DOI:** 10.3390/vaccines11030677

**Published:** 2023-03-16

**Authors:** Vivek Kapur

**Affiliations:** Department of Animal Sciences, The Huck Institutes of the Life Sciences, The Pennsylvania State University, University Park, PA 16802, USA; vkapur@psu.edu

I read with interest the article “A Cost-Effectiveness Analysis of Pre-Exposure Prophylaxis to Avert Rabies Deaths in School-Aged Children in India” by Royal et al., recently published in *Vaccines* [1].

Given the reported high burden of morbidity and mortality associated with rabies in India [2,3,4], rigorous assessments of public health interventions to reduce disease burden in especially vulnerable populations are much needed and are welcome. 

To this end, Royal and colleagues developed a decision-tree model to assess the cost-effectiveness of several pre-exposure (PrEP) and post-exposure prophylaxis (PEP) strategies to avert rabies mortalities in school-aged (5–15-year-old) children in India [1]. 

The results of their analyses suggest that a two-site intradermal PrEP regimen administered on days zero and seven is “very cost effective”, as opposed to eight comparator regimens, with up to 89.9% fewer deaths resulting from the implementation of the proposed PrEP intervention over the standard of care, per the national guidelines. Based on incremental costs, the authors conclude that PrEP is a cost-effective intervention and will reduce rabies-associated mortality and morbidity in the more than 230 million at-risk school-aged children in India [1]. 

However, a closer examination of the results raises major concerns regarding the validity of both the model and the conclusions. 

First, the model presented by Royal and colleagues predict orders of magnitude greater that the deaths in India from rabies than are reported based on surveys by the WHO or by multiple independent surveys and model-based estimates. For instance, an oft-cited WHO-sponsored multi-centric survey from 2003 reports ~20 deaths per million per annum from rabies across all age groups within India [2]; the Indian Govt. recorded fewer than 1 death per million in 2019 [5]; and a recent study based on data from the Global Burden of Diseases (GBD) 2019 survey estimated ~4 deaths per million [4]. In stark contrast, the model developed by Royal et al. predicted that their favored PrEP regimen (pre-exposure prophylaxis plus post-exposure prophylaxis, per the WHO guidelines) would result in 273 deaths per million of the population (see Table 3, in [1]). This is an order of magnitude greater the estimated annual rabies deaths per the WHO and exceeds by three or more orders of magnitude the Govt. of India’s estimates. Of perhaps greater concern, the Royal model predicts that comparator post-exposure prophylaxis-only regimens that include the current standard of care would result in, on average, greater than 2500 deaths per million [1]. If accurate, this extraordinarily large number of predicted rabies deaths would not only be multiple orders of magnitude above the current independent estimates of rabies mortality in India, but would implausibly be twice as large as the annual sum of deaths resulting from all other communicable diseases across all age groups in the nation [5].

This unexplainedly high number of predicted deaths and number of deaths averted by the Royal et al. model is of critical relevance to the central hypothesis of the study since the cost-effectiveness of interventions for invariably fatal diseases such as rabies is driven largely by the years of lives lost (derived as a product of the number of deaths and the life expectancy beyond age of death), and cost-effectiveness estimates are highly sensitive to the number of predicted deaths amongst the comparator intervention arms. Thus, given the implausibly high orders of magnitude greater estimates of predicted deaths by the Royal et al. model as compared with the prevailing evidence from multiple independent reports, including from the WHO and official Govt. of India notifications, the predicted estimates of the incremental cost-effectiveness ratio (ICER) are likely to be flawed.

Second, the authors have noted that key model data inputs, such as the probability of a dog bite, and the probability of a bite resulting from a rabid animal, as well as various other parameters, were extracted from the published estimates and assumptions validated by field experts (see Tables 2 and A1; [1]). While this approach is perfectly justifiable, the model’s outputs, nonetheless, appear to predict improbably large numbers of rabies-related deaths in India. Why and how might this occur? Curious readers perhaps might not have to look far since a recent systematic review of dog-mediated rabies in India (performed by some of the current study’s authors) has already highlighted the dearth of nationally representative and generalizable studies and estimates, as well as the general low quality of published evidence surrounding rabies-related and considerable associated uncertainties [6]. 

It is well understood that the parameterization of models with a poor quality of evidence from non-representative or generalizable studies often results in outputs of a questionable validity. For instance, in the model developed by Royal et al., the base rate for the probability of a dog bite was taken to be 4.4% per annum [1]. Their estimate is derived from a single cross-sectional investigation performed in an urban setting (Patna, India) that noted 46 dog bites amongst a surveyed population of 1045 5-to-14-year-old children over a 13 month period [7]. However, Patna’s study was not designed to be nationally representative, and neither was it claimed by the original study’s authors to be representative or generalizable for the entire country. In contrast, a nationally representative WHO-sponsored survey of more than 50,000 dog-bite victims distributed across rural and urban settings in 18 States in India suggested a 2.5% annual incidence rate in children less than 14 years of age [8]. When considered in the context of the more than 230 million children in this age cohort, the higher bite rate selected by Royal et al. would result in 75% (or 4 million) excess cases each year, contributing, in part, to the extraordinarily high rabies-related death rates predicted by their model. 

Perhaps of a greater consequence, Royal et al. assumed a probability of a bite being from a rabid animal of 29.5% [1]. This seems extraordinarily high, with the estimate apparently derived from a report of a WHO-sponsored multi-centric survey of rabies in India performed in 2017 [9]. However, in Section 3.3.2.2, the survey notes that “…29.5% of the biting animals showed some signs of suspected rabies such as aggression, hypersalivation, biting other animals and changes in dog bark; but none of the biting animal (sic) was proven to be rabid (Table 18)” (emphasis added) [9]. Given that none of the ~150 biting animals recorded in this survey with clinical signs were confirmed to be rabid, the assumption by Royal et al. that all 29.5% are rabid does not seem justifiable. Even in a country in which the laboratory confirmation of rabies is rare, to consider that every biting animal that exhibits the highly non-specific sign of “aggression” is rabid seems ill-advised. This is especially so since only 1% or fewer of these same animals exhibited other rabies-associated signs such hyper-salivation or a change in bark (Table 18; [9]). Thus, the application of this single parameter estimate, derived from low-quality and misinterpreted evidence, likely contributes to the orders of magnitude greater predicted rabies deaths in India by Royal et al. [1].

Taken together, despite the assertions that the newly developed model is parameterized with estimates from published sources and has been vetted by field experts, it outputs implausibly high rates of rabies-related deaths, for which neither a rationale is provided, and nor are the implications of this anomalous finding discussed by the authors. This omission is noteworthy since a higher burden of proof is perhaps required prior to consideration of PrEP as an intervention to prevent potentially phantom rabies-related deaths amongst the 230 million school-aged children in India, given that such an endeavor may instead end up stretching an already overburdened and chronically under-resourced public health system past its breaking point. Hence, the conclusion by Royal et al. that “…the PrEP (I) regimen is a cost-effective and life-saving strategy to avert painful mortalities due to rabies in school-aged children in India” appears at best to be premature, and the null hypothesis should not be rejected based on the presented evidence. 

Every single death from rabies represents a tragic loss. Hence, there is undoubtedly a real and unmet need for renewed efforts to mitigate this preventable loss through an empirical and model-based assessment of interventions, such as those evaluated in this study. The shortcomings associated with the poor quality of evidence in the Royal et al. study shows precisely why it is high time for some clear-eyed thinking regarding the control of rabies in India, and why perhaps it is time to take a step or two back and seek a first-principles approach to obtaining robust and reliable estimates from which to parameterize models such as these. In this context, the recent launch of the national action plan for dog-mediated rabies elimination from India by 2030, and the declaration of rabies as a notifiable disease in India, appears to be a step in the right direction [10].

In conclusion, the helpful aphorism attributed to George Box that “all models are wrong, but some may be useful” [11] is worth remembering in the context of the study by Royal et al. Unfortunately, given the orders of magnitude differences in the estimates of rabies-related death rates (and costs) compared with multiple prior independent investigations and official Government reports, it appears that the model developed by Royal et al. is likely to be wrong, and its usefulness is perhaps limited to highlighting the urgent and unmet need to fill in key knowledge gaps regarding the burden and risks associated with rabies-related mortality and morbidity in India.

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
