# Peer review of "Is Pre-Exposure Prophylaxis a Cost-Effective Intervention to Avert Rabies Deaths among School-Aged Children in India? Comment on Royal et al. A Cost-Effectiveness Analysis of Pre-Exposure Prophylaxis to Avert Rabies Deaths in School-Aged Children in India. Vaccines 2023, 11, 88"

_vaccines, 2023, doi:10.3390/vaccines11030677_

Round 1

Reviewer 1 Report

I read with interest the comment of VK to the recently published article entitled “Pre-exposure Prophylaxis a Cost-effective Intervention to Avert Rabies Deaths in School-Aged Children in India”.

Given the limited availability or lack of reliable data on rabies cases in India, that is well-recognized by both the Authors and VK, she criticized the approach adopted by the Authors, the model definition and the conclusions.

It seems that the proposed model has no chance to be accepted now and in the future.

In their reply, the Authors review the main point of the article and give explanations supported by selected references remarking the limitations inherent the model.

A divergence does exist on the reference literature (Sudarshan et al., 2006) concerning surveys previously carried out in India to assess the burden of human rabies in terms of reliability of the articles cited due to the time when the survey was carried out, range of age considered and territorial approach.

In my opinion, both the comment of VK “Is Pre-exposure Prophylaxis a Cost-effective Intervention to Avert Rabies Deaths in School-Aged Children in India?” and the letter to Editor by Royal et al. deserve to be published.

Rabies is clearly a severe matter of concern for both human and animal health in India and any attempt to draw attention to this topic is relevant. This is true also for a modelling evaluation considering pre-exposure prophylaxis and its cost-effectiveness, despite the limited availability or lack of reliable data.

Author Response

Thank you for your thoughtful and considered review of the Letter to the Editor and the response of Royal et al. and the recommendation to publish both.  

I fully concur that rabies is an important challenge to both human and animal health in India and that approaches, including modeling and field based assessments of of cost-effective solutions to address this are needed even while major data gaps are being filled, and hopefully this will be highlighted by publication of the Letter to the Editor and the author' response.

Reviewer 2 Report

Much of the critiscism lies in the data collected. The author used data from the Indian government as a better source. Is the Indian government a better source than say WHO? Data coming from third world countries are often unreliable. What is more important in the paper by Royal et ak is the methodology provided, not necessarily the data inputed or the the results,. Most readers will take the results of Royal et al as a pinch of salt just ;ikle virtually all papers.

The letter does not provide sufficient challenge to the paper..

Author Response

Thank you for the review and comments.  

The primary criticism of the Royal et al study raised in the Letter to the Editors does not lie in the data collected, and nor does it vouch for or speak to the accuracy of either Government of India data or WHO data.  The letter does not as well question the general methodology that was applied by Royal et al. since only standard and previously published models were used, this part was not novel.  The crux of the criticism of the Royal et al study as also highlighted in the letter is that, no matter the source of input data used to parameterize, their outputs an implausible 10 to 1000 or more times greater rabies mortality than reported by ANY source (Govt. of India or WHO or other), with numbers of predicted rabies related deaths under status quo that perhaps exceed all causes of mortality in the target age group population.  These numbers are not mere quibbles to be taken with a "pinch of salt" as the reviewer suggests, but go to the very heart of the conclusions of the Royal et al study since it is the numbers of deaths saved that drives the apparent cost-efficiency of the pre-exposure prophylaxis claimed by the Royal et al model.  If the numbers of deaths are lower, the cost-efficiency is likely to disappear.  This is a fundamental challenge to the conclusions of the paper.

Since this extraordinarily discrepant and critical output that results from their model was neither noted nor discussed by Royal et al in their publication, the Letter to the Editors attempts to highlight this questionable finding, identify the potential sources of the error, and highlights major data gaps relating to rabies in India for further study in a manner that might be of considerable interest to the readers of Vaccines.

Finally, I respectfully note that India is NOT a "third-world country" as the reviewer implies.  India has the fourth largest economy in the world, supplies the world with most of its vaccines and medications, is the brains behind a large proportion of the world's software, and boasts a robust (though chronically underfunded) public and private health care system. India just happens to be classified by the World Bank as a lower middle income country (https://data.worldbank.org/?locations=IN-XN), and hence should be referred to as such when an appropriate context might arise.